# LEARNING STOCHASTIC BEHAVIOUR FROM AGGREGATE DATA

## ABSTRACT

Learning nonlinear dynamics from aggregate data is a challenging problem since the full trajectory of each individual is not available, namely, the individual observed at one time point may not be observed at next time point, or the identity of individual is unavailable. This is in sharp contrast to learning dynamics with trajectory data, on which the majority of existing methods are based. We propose a novel method using the weak form of Fokker Planck Equation (FPE) to describe density evolution of data in a sampling form, which is then combined with Wasserstein generative adversarial network (WGAN) in training process. In such a sample-based framework we are able to study nonlinear dynamics from aggregate data without solving the partial differential equation (PDE). The model can also handle high dimensional cases with the help of deep neural networks. We demonstrate our approach in the context of a series of synthetic and real-world data sets.

## 1 INTRODUCTION

In the context of a dynamic system, **Aggregate data** refers to the data sets that full trajectory of each individual is not available, meaning that there is no known individual level correspondence. Typical examples include data sets collected for DNA evolution, social gathering, density in control problems, bird migration during which it is impossible to identify individual bird, and many more. In those applications, some observed individuals at one time point may be unobserved at the next time spot, or when the individual identities are blocked or unavailable due to various technical and ethical reasons. Rather than inferring the exact information for each individual, the main objective of learning dynamics in aggregate data is to recover and predict the evolution of distribution of all individuals together. **Trajectory data**, in contrast, is a kind of data that we are able to acquire the information of each individual all the time, although some studies considered the case that some individual trajectories are partially missing. However, the identities of those individuals, whenever they are observable, is always assumed available. For example, stock price, weather, customer behaviors and most training data sets for computer vision and natural language processing.

There are many popular models to learn dynamics of full-trajectory data. Typical ones include Hidden Markov Model (HMM)(Alshamaa et al., 2019; Eddy, 1996), Kalman Filter (KF)(Farahi & Yazdi, 2020; Harvey, 1990; Kalman, 1960) and Particle Filter (PF) (Santos et al., 2019; Djuric et al., 2003), as well as the models built upon HMM, KF and PF(Deriche et al., 2020; Fang et al., 2019; Hefny et al., 2015; Langford et al., 2009), they all require full trajectories of each individual, which may not be applicable in the aggregate data situations. On the other side, only a few methods are focused on aggregated data in the recent learning literature. In the work of Hashimoto et al. (2016), authors assumed that the hidden dynamic of particles follows a stochastic differential equation(SDE), in particular, they use a recurrent neural network to parameterize the drift term. Furthermore, Wang et al. (2018) improved traditional HMM model by using an SDE to describe the evolving process of hidden states. To the best of our knowledge, there is no method directly learning the evolution of the density of objects from aggregate data yet.

We propose to learn the dynamics of density through the weak form of Fokker Planck Equation (FPE), which is a parabolic partial differential equation (PDE) governing many dynamical systems subject to random noise perturbations, including the typical SDE models in existing studies. Our learning is accomplished by minimizing the Wasserstein distance between predicted distribution given by FPE and the empirical distribution from data samples. Meanwhile we utilize neural networks to handle higher dimensional cases. More importantly, by leveraging the framework of Wasserstein Generative

Adversarial Network (WGAN) (Arjovsky et al., 2017), our model is capable of approximating the distribution of samples at different time points without solving the SDE or FPE. More specifically, we treat the drift coefficient, the goal of learning, in the FPE as a generator, and the test function in the weak form of FPE as a discriminator. In other words, our method can also be regarded as a data-driven method to estimate transport coefficient in FPE, which corresponds to the drift terms in SDEs. Additionally, though we treat diffusion term as a constant in our model, it is straightforward to generalize it to be a neural network as well, which can be an extension of this work. We would like to mention that several methods of solving SDE and FPE (Weinan et al., 2017; Beck et al., 2018; Li et al., 2019) adopt opposite ways to our method, they utilize neural networks to estimate the distribution $P(x, t)$ with given drift and diffusion terms.

In conclusion, our contributions are:

- We design an algorithm that is able to recover the density evolution of nonlinear dynamics via minimizing the Wasserstein discrepancy between real aggregate data and our generated data.

- By leveraging the weak form of FPE, we are able to compute the Wasserstein distance directly without solving the FPE.

- Finally, we demonstrate the accuracy and the effectiveness of our algorithm by several synthetic and real-world examples.

## 2 PROPOSED METHOD

### 2.1 FOKKER PLANCK EQUATION FOR THE DENSITY EVOLUTION

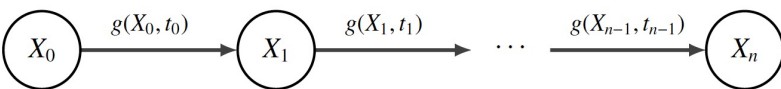

Figure 1: State model of the stochastic process $X_t$

We assume the individuals evolve in a pattern in the space $\mathbb{R}^D$ as shown in Figure 1. One example satisfying such process is the stochastic differential equation(SDE), which is also known as the Itô process (Øksendal, 2003): $d\boldsymbol{X}_t = g(\boldsymbol{X}_t, t)dt + \sigma d\boldsymbol{W}_t$. Here $d\boldsymbol{X}_t$ represents an infinitesimal change of $\{\boldsymbol{X}_t\}$ along with time increment $dt$, $g(\cdot, t) = (g^1(\cdot, t), ..., g^D(\cdot, t))^T$ is the drift term (drifting vector field) that drives the dynamics of the SDE, $\sigma$ is the diffusion constant, $\{\boldsymbol{W}_t\}$ is the standard Brownian Motion. Then the probability density of $\{\boldsymbol{X}_t\}$ is governed by the Fokker Planck Equation(FPE) (Risken & Caugheyz, 1991), as stated below in Lemma 1.

**Lemma 1.** *Suppose $\{\boldsymbol{X}_t\}$ solves the SDE $d\boldsymbol{X}_t = g(\boldsymbol{X}_t, t)dt + \sigma d\boldsymbol{W}_t$, denote $p(\cdot, t)$ as the probability density of the random variable $\boldsymbol{X}_t$. Then $p(x, t)$ solves the following equation:*

$$\frac{\partial p(\boldsymbol{x}, t)}{\partial t} = \sum_{i=1}^{D} -\frac{\partial}{\partial x_i} \left[ g^i(\boldsymbol{x}, t) p(\boldsymbol{x}, t) \right] + \frac{1}{2}\sigma^2 \sum_{i=1}^{D} \frac{\partial^2}{\partial x_i^2} p(\boldsymbol{x}, t) \tag{1}$$

As a linear evolution PDE, FPE describes the evolution of density function of the stochastic process driven by a SDE. Due to this reason, FPE plays a crucial role in stochastic calculus, statistical physics and modeling (Nelson, 1985; Qi & Majda, 2016; Risken, 1989). Its importance is also drawing more attention among statistic and machine learning communities (Liu & Wang, 2016; Pavon et al., 2018; Rezende & Mohamed, 2015). In this paper, we utilize the weak form of FPE as a basis to study hidden dynamics of the time evolving aggregated data without solving FPE.

Our task can be described as: assume that the individuals evolve with the process indicated by Figure 1, which can be simulated by Itô process. Then given observations $\boldsymbol{x}_t$ along time axis, we aim to recover the drift coefficient $g(\boldsymbol{x}, t)$ in FPE, and thus we are able to recover and predict the density evolution of such dynamic. For simplicity we treat $g(\boldsymbol{x}, t)$ as a function uncorrelated to time $t$, namely,

$g(\boldsymbol{x}, t) = g(\boldsymbol{x})$. Notice that though evolving process of individuals can be simulated by Itô process, in reality since we lose identity information of individuals, the observed data become aggregate data, thus we need a new way other than traditional methods to study the swarm's distribution.

## 2.2 Weak Form of Fokker Planck Equation

Given FPE stated in Lemma 1, if we multiply a test function $f \in H_0^1(\mathbb{R}^D)$ on both sides of the FPE, then the integration on both sides leads to:

$$\int \frac{\partial p}{\partial t} f(\boldsymbol{x}) d\boldsymbol{x} = \int \sum_{i=1}^{D} -\frac{\partial}{\partial x_i} \left[ g^i(\boldsymbol{x}) p(\boldsymbol{x}, t) \right] f(\boldsymbol{x}) d\boldsymbol{x} + \frac{1}{2} \sigma^2 \int \sum_{i=1}^{D} \frac{\partial^2}{\partial x_i^2} p(\boldsymbol{x}, t) f(\boldsymbol{x}) d\boldsymbol{x}.$$

Then integrating by parts on the right hand side leads to the weak form of FPE:

$$\int \frac{\partial p}{\partial t} f(\boldsymbol{x}) d\boldsymbol{x} = \int \sum_{i=1}^{D} g^i(\boldsymbol{x}) \frac{\partial}{\partial x_i} f(\boldsymbol{x}) p(\boldsymbol{x}, t) d\boldsymbol{x} + \frac{1}{2} \sigma^2 \int \sum_{i=1}^{D} \frac{\partial^2}{\partial x_i^2} f(\boldsymbol{x}) p(\boldsymbol{x}, t) d\boldsymbol{x}.$$

where $H_0^1(\mathbb{R}^D)$ denote the Sobolev space. The first advantage of weak solution is that the solution of a PDE usually requires strong regularity and thus may not exist in the classical sense for a certain group of equations, however, the weak solution has fewer regularity requirements and thus their existence are guaranteed for a much larger classes of equations. The second advantage is that the weak formulation may provide new perspectives for numerically solving PDEs (Zienkiewicz & Cheung, 1971),(Sirignano & Spiliopoulos, 2018) and (Zang et al., 2019) etc.

Suppose the observed samples at time points $t_{m-1}$ and $t_m$ are following true densities $\hat{p}(\cdot, t_{m-1})$ and $\hat{p}(\cdot, t_m)$. Consider the following SDE

$$d\tilde{\boldsymbol{X}}_t = g_\omega(\tilde{\boldsymbol{X}}_t) dt + \sigma d\boldsymbol{W}_t, \quad t_{m-1} \le t \le t_m \quad \tilde{\boldsymbol{X}}_{t_{m-1}} \sim \hat{p}(\cdot, t_{m-1}) \tag{2}$$

Here $g_\omega$ is an approximation to the real drift term $g$, in our research, we treat $g_\omega$ as a neural network with parameters $\omega$. Let us now denote $\tilde{p}(\cdot, t)$ as the density of $\tilde{\boldsymbol{X}}_t$.

Then it is natural to compute and minimize the discrepancy between our approximated density $\tilde{p}(\cdot, t_m)$ and true density $\hat{p}(\cdot, t_m)$, within this process, we are optimizing $g_\omega$ and thus will recover the true drift term $g$. In our research, we choose the Wasserstein-1 distance as our discrepancy function (Villani, 2008) (Arjovsky et al., 2017). Now we apply Kantorovich-Rubinstein duality (Villani, 2008), this leads to:

$$W_1(\hat{p}(\cdot, t_m), \tilde{p}(\cdot, t_m)) = \sup_{\|\nabla f\| \le 1} \left\{ \mathbb{E}_{\boldsymbol{x}_r \sim \hat{p}(\boldsymbol{x}, t_m)}[f(\boldsymbol{x}_r)] - \mathbb{E}_{\boldsymbol{x}_g \sim \tilde{p}(\boldsymbol{x}, t_m)}[f(\boldsymbol{x}_g)] \right\} \tag{3}$$

The first term $\mathbb{E}_{\boldsymbol{x}_r \sim \hat{p}(\boldsymbol{x}, t_m)}[f(\boldsymbol{x}_r)]$ in Equation (3) can be conveniently computed by Monte-Carlo method since we are already provided with the real data points $\boldsymbol{x}_r \sim \hat{p}(\cdot, t_m)$. To evaluate the second term, we first approximate $\tilde{p}(\cdot, t_m)$ by trapezoidal rule (Atkinson, 2008):

$$\tilde{p}(\boldsymbol{x}, t_m) \approx \hat{p}(\boldsymbol{x}, t_{m-1}) + \frac{\Delta t}{2} \left( \frac{\partial \tilde{p}(\boldsymbol{x}, t_{m-1})}{\partial t} + \frac{\partial \tilde{p}(\boldsymbol{x}, t_m)}{\partial t} \right) \quad \text{here } \Delta t = t_m - t_{m-1}. \tag{4}$$

And thus we can compute:

$$\mathbb{E}_{\boldsymbol{x}_g \sim \tilde{p}(\cdot, t_m)}[f(\boldsymbol{x}_g)] \approx \int f(\boldsymbol{x}) \hat{p}(\boldsymbol{x}, t_{m-1}) d\boldsymbol{x} + \frac{\Delta t}{2} \left( \int \frac{\partial \hat{p}(\boldsymbol{x}, t_{m-1})}{\partial t} f(\boldsymbol{x}) d\boldsymbol{x} + \int \frac{\partial \tilde{p}(\boldsymbol{x}, t_m)}{\partial t} f(\boldsymbol{x}) d\boldsymbol{x} \right) \tag{5}$$

In the above Equation (5), the second and the third term on the right-hand side can be reformulated via the weak form of FPE and we are able to derive a computable formulation for $W_1(\hat{p}(\cdot, t_m), \tilde{p}(\cdot, t_m))$. Furthermore, we can use Monte-Carlo method to approximate the expectations in (5). the first and the second terms can be directly computed via data points from $\hat{p}(\cdot, t_{m-1})$. For the third term, we need to generate samples from $\tilde{p}(\cdot, t_m)$, to achieve this, we apply Euler-Maruyama scheme (Kloeden & Platen, 2013) to SDE (2) in order to acquire our desired samples $\tilde{\boldsymbol{x}}_{t_m}$:

$$\tilde{\boldsymbol{x}}_{t_m} = \hat{\boldsymbol{x}}_{t_{m-1}} + g_\omega(\hat{\boldsymbol{x}}_{t_{m-1}}) \Delta t + \sigma \sqrt{\Delta t} z, \quad \text{here } z \sim \mathcal{N}(0, I), \quad \hat{\boldsymbol{x}}_{t_{m-1}} \sim \hat{p}(\cdot, t_{m-1}). \tag{6}$$

Here $\mathcal{N}(0, I)$ is the standard Gaussian distribution on $\mathbb{R}^D$. Now we summarize these results in Proposition 1:

**Proposition 1.** *For a set of points $X = \{x^{(1)}, ..., x^{(N)}\}$ in $\mathbb{R}^D$. We denote:*

$$\mathcal{F}_f(X) = \frac{1}{N} \sum_{k=1}^{N} \left( \sum_{i=1}^{D} g_\omega^i(x^{(k)}) \frac{\partial}{\partial x_i} f(x^{(k)}) + \frac{1}{2}\sigma^2 \sum_{i=1}^{D} \frac{\partial^2}{\partial x_i^2} f(x^{(k)}) \right)$$

*then at time point $t_m$, the Wasserstein distance between $\hat{p}(\cdot, t_m)$ and $\tilde{p}(\cdot, t_m)$ can be approximated by:*

$$W_1(\hat{p}(\cdot, t_m), \tilde{p}(\cdot, t_m)) \approx \sup_{\|\nabla f\| \leq 1} \left\{ \frac{1}{N} \sum_{k=1}^{N} f(\hat{x}_{t_m}^{(k)}) - \frac{1}{N} \sum_{k=1}^{N} f(\hat{x}_{t_{m-1}}^{(k)}) - \frac{\Delta t}{2}(\mathcal{F}_f(\hat{X}_{m-1}) + \mathcal{F}_f(\tilde{X}_m)) \right\}$$

*Here $\{\hat{x}_{t_{m-1}}^{(k)}\} \sim \hat{p}(\cdot, t_{m-1})$, $\{\hat{x}_{t_m}^{(k)}\} \sim \hat{p}(\cdot, t_m)$ and we denote $\hat{X}_{m-1} = \{\hat{x}_{t_{m-1}}^{(1)}, ..., \hat{x}_{t_{m-1}}^{(N)}\}$, $\tilde{X}_m = \{\tilde{x}_{t_m}^{(1)}, ..., \tilde{x}_{t_m}^{(N)}\}$, where each $\tilde{x}_{t_m}^{(k)}$ is computed by Euler-Maruyama scheme (6).*

## 2.3 Wasserstein Distance on Time Series

In real cases, it is not realistic to observe the data at arbitrary two consecutive time nodes, especially when $\Delta t$ is small. To make our model more flexible, we should extend our formulation so that we are able to plug in observed data at arbitrary time points. To be more precise, suppose we observe data set $\hat{X}_{t_n} = \{\hat{x}_{t_n}^{(1)}, ..., \hat{x}_{t_n}^{(N)}\}$ at $J + 1$ different time points $t_0, t_1, ..., t_J$. And we denote the generated data set as $\tilde{X}_{t_n} = \{\tilde{x}_{t_n}^{(1)}, ..., \tilde{x}_{t_n}^{(N)}\}$, here each $\tilde{x}_{t_n}^{(\cdot)}$ is derived from the $n$-step Euler-Maruyama scheme:

$$\tilde{x}_{t_j} = \tilde{x}_{t_{j-1}} + g_\omega(\tilde{x}_{t_{j-1}})\Delta t + \sigma\sqrt{\Delta t}z, \quad \text{with } z \sim \mathcal{N}(0, I) \quad 0 \leq j \leq n, \quad \tilde{x}_{t_0} \sim \hat{p}(\cdot, t_0) \qquad (7)$$

Let us denote $\tilde{p}(\cdot, t)$ as the solution to FPE (1) with $g$ replaced by $g_\omega$ and with initial condition $\tilde{p}(\cdot, t_0) = \hat{p}(\cdot, t_0)$, then the approximation formula for evaluating the Wasserstein distance $W_1(\hat{p}(\cdot, t_n), \tilde{p}(\cdot, t_n))$ is provided in the following proposition:

**Proposition 2.** *Suppose we keep all the notations defined as above, then we have the approximation:*

$$W_1(\hat{p}(\cdot, t_n), \tilde{p}(\cdot, t_n)) \approx \sup_{\|\nabla f\| \leq 1} \left\{ \frac{1}{N} \sum_{k=1}^{N} f(\hat{x}_{t_n}^{(k)}) - \frac{1}{N} \sum_{k=1}^{N} f(\hat{x}_{t_0}^{(k)}) - \frac{\Delta t}{2}(\mathcal{F}_f(\hat{X}_0) + \mathcal{F}_f(\tilde{X}_n) + 2\sum_{s=1}^{n-1} \mathcal{F}_f(\tilde{X}_s)) \right\}$$

**Minimizing the Objective Function:** Based on Proposition 2, we obtain objective function by summing up the accumulated Wasserstein distances among $J$ observations along the time axis. Thus, our ultimate goal is to minimize the following objection function:

$$\min_{g_\omega} \left\{ \sum_{n=1}^{J} \sup_{\|\nabla f_n\| \leq 1} \left\{ \frac{1}{N} \sum_{k=1}^{N} f_n(\hat{x}_{t_n}^{(k)}) - \frac{1}{N} \sum_{k=1}^{N} f_n(\hat{x}_{t_0}^{(k)}) - \frac{\Delta t}{2}(\mathcal{F}_{f_n}(\hat{X}_0) + \mathcal{F}_{f_n}(\tilde{X}_n) + 2\sum_{s=1}^{n-1} \mathcal{F}_{f_n}(\tilde{X}_s)) \right\} \right\}$$

Notice that since we have observations on $J$ distinct time points(despite the initial point), for each time point we compute Wasserstein distance with the help of the dual function $f_n$, thus we will involve $J$ test functions in total. In our actual implementation, we will choose these dual functions as neural networks. We call our algorithm Fokker Planck Process(FPP), the entire procedure is shown in Algorithm 1. We also provide an error analysis in Appendix.

## 3 Experiments

In this section, we evaluate our model on various synthetic and realistic data sets by employing Algorithm 1. We generate samples $\tilde{x}_t$ and make all predictions base on Equation (6) starting with $\hat{x}_0$.

**Baselines:** We compare our model with two recently proposed methods. One model (NN) adopts recurrent neural network(RNN) to learn dynamics directly from observations of aggregate data (Hashimoto et al., 2016). The other one model (LEGEND) learns dynamics in a HMM framework (Wang et al., 2018). The baselines in our experiments are two typical representatives that have state-of-the-art performance on learning aggregate data. Furthermore, though we simulate the evolving process of the data as a SDE, which is on the same track with NN, as mentioned before, NN trains its RNN via optimizing Sinkhorn distance (Cuturi, 2013), our model starts with a view of weak form of PDE, focuses more on WGAN framework and easier computation.

---

**Algorithm 1** Fokker Planck Process Algorithm

---

**Require:** Initialize $f_{\theta_n}$ $(1 \le n \le J)$, $g_\omega$
**Require:** Set $\epsilon_{f_n}$ as the inner loop learning rate for $f_{\theta_n}$ and $\epsilon_g$ as the outer loop learning rate for $g_\omega$
 1: **for** # training iterations **do**
 2:     **for** k steps **do**
 3:         **for** observed time $t_s$ in $\{t_1, ..., t_J\}$ **do**
 4:             Compute the generated data set $\tilde{X}_{t_s}$ from Euler-Maruyama scheme (7) for $1 \le s \le J$
 5:             Acquire data sets $\hat{X}_{t_s} = \{\hat{\boldsymbol{x}}_{t_s}^{(1)}, ..., \hat{\boldsymbol{x}}_{t_s}^{(N)}\}$ from real distribution $\hat{p}(\cdot, t_s)$ for $1 \le s \le J$
 6:         **end for**
 7:         For each dual function $f_{\theta_n}$, compute:
 8:         $\mathcal{F}_n = \mathcal{F}_{f_{\theta_n}}(\hat{X}_{t_0}) + \mathcal{F}_{f_{\theta_n}}(\tilde{X}_{t_n}) + 2 \sum_{s=1}^{n-1} \mathcal{F}_{f_{\theta_n}}(\tilde{X}_{t_s})$
 9:         Update each $f_{\theta_n}$ by $\theta_n \leftarrow \theta_n + \epsilon_{f_n} \nabla_\theta \left( \frac{1}{N} \sum_{k=1}^N f_{\theta_n}(\hat{\boldsymbol{x}}_{t_n}^{(k)}) - \frac{1}{N} \sum_{k=1}^N f_{\theta_n}(\hat{\boldsymbol{x}}_{t_0}^{(k)}) - \frac{\Delta t}{2} \mathcal{F}_n \right)$
10:     **end for**
11:     Update $g_\omega$ by $\omega \leftarrow \omega - \epsilon_g \nabla_\omega \left( \sum_{n=1}^J (\frac{1}{N} f_{\theta_n}(\hat{\boldsymbol{x}}_{t_n}^{(k)}) - \frac{1}{N} f_{\theta_n}(\hat{\boldsymbol{x}}_{t_0}^{(k)}) - \frac{\Delta t}{2} \mathcal{F}_n) \right)$
12: **end for**

---

### 3.1 SYNTHETIC DATA

We first evaluate our model on three synthetic data sets which are generated by three artificial dynamics: **Synthetic-1**, **Synthetic-2** and **Synthetic-3**.

**Experiment Setup:** In all synthetic data experiments, we set the drift term $g$ and the discriminator $f$ as two simple fully-connected networks. The $g$ network has one hidden layer and the $f$ network has three hidden layers. Each layer has 32 nodes for both $g$ and $f$. The only one activation function we choose is Tanh. Notice that since we need to calculate $\frac{\partial^2 f}{\partial x^2}$, the activation function of $f$ must be twice differentiable to avoid loss of weight gradient. In terms of training process, we use the Adam optimizer (Kingma & Ba, 2014) with learning rate $10^{-4}$. Furthermore, we use spectral normalization to realize $\|\nabla f\| \le 1$(Miyato et al., 2018). We initialize the weights with Xavier initialization(Glorot & Bengio, 2010) and train our model by Algorithm 1.

**Synthetic-1:**

$$\hat{\boldsymbol{x}}_0 \sim \mathcal{N}(0, \Sigma_0), \qquad \hat{\boldsymbol{x}}_{t+\Delta t} = \hat{\boldsymbol{x}}_t - (\boldsymbol{A}\hat{\boldsymbol{x}}_t + \boldsymbol{b})\Delta t + \sigma \sqrt{\Delta t}\mathcal{N}(0, 1)$$

**Synthetic-2:**

$$\hat{\boldsymbol{x}}_0 \sim \mathcal{N}(0, \Sigma_0), \qquad \hat{\boldsymbol{x}}_{t+\Delta t} = \hat{\boldsymbol{x}}_t - \boldsymbol{G}\Delta t + \sigma \sqrt{\Delta t}\mathcal{N}(0, 1)$$

$$\boldsymbol{G} = \begin{bmatrix} \frac{1}{\sigma_1} \frac{N_1}{N_1+N_2}(\hat{x}_t^1 - \mu_{11}) + \frac{1}{\sigma_2} \frac{N_2}{N_1+N_2}(\hat{x}_t^1 - \mu_{21}) & 0 \\ 0 & \frac{1}{\sigma_1} \frac{N_1}{N_1+N_2}(\hat{x}_t^2 - \mu_{12}) + \frac{1}{\sigma_2} \frac{N_2}{N_1+N_2}(\hat{x}_t^2 - \mu_{22}) \end{bmatrix}$$

$$N_1 = \frac{1}{\sqrt{2\pi}\sigma_1} \exp\left(-\frac{(\hat{x}_t^1 - \mu_{11})^2}{2\sigma_1^2} - \frac{(\hat{x}_t^1 - \mu_{12})^2}{2\sigma_1^2}\right), \quad N_2 = \frac{1}{\sqrt{2\pi}\sigma_2} \exp\left(-\frac{(\hat{x}_t^2 - \mu_{21})^2}{2\sigma_2^2} - \frac{(\hat{x}_t^2 - \mu_{22})^2}{2\sigma_2^2}\right)$$

**Synthetic-3 (Van der Pol oscillator (Li, 2018)):**

$$\hat{\boldsymbol{x}}_0 \sim \mathcal{N}(0, \Sigma_0)$$

$$\hat{x}_{t+\Delta t}^1 = \hat{x}_t^1 + 10\left(\hat{x}_t^2 - \frac{1}{3}(\hat{x}_t^1)^3 + \hat{x}_t^1\right)\Delta t + \sigma \sqrt{\Delta t}\mathcal{N}(0, 1)$$

$$\hat{x}_{t+\Delta t}^2 = \hat{x}_t^2 + 3(1 - \hat{x}_t^1)\Delta t + \sigma \sqrt{\Delta t}\mathcal{N}(0, 1)$$

The data size at each time point is $N = 2000$, the dimension of the data is $D = 2$. We treat 1200 data points as the training set and the other 800 data points as the test set. In Synthetic-1, the data is following a simple linear dynamic, where $\boldsymbol{A} = [(4, 0), (0, 1)], \boldsymbol{b} = [-12, -12]^T$. We let $\Delta t = 0.01$, $\sigma = 1, \Sigma_0 = [(1, 0), (0, 1)]$. We utilize true $\boldsymbol{x}_0$, $\boldsymbol{x}_{20}$ and $\boldsymbol{x}_{200}$ in training process and predict the

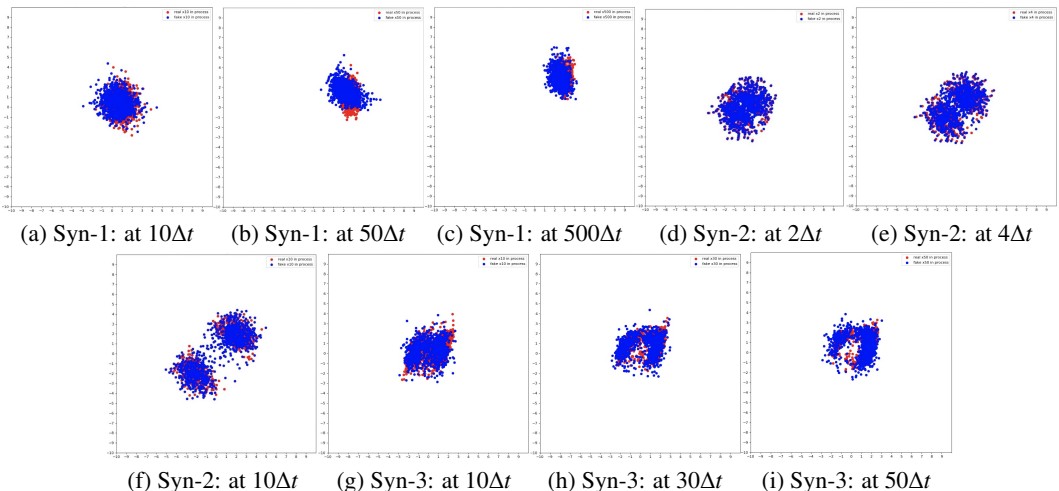

(a) Syn-1: at $10\Delta t$  (b) Syn-1: at $50\Delta t$  (c) Syn-1: at $500\Delta t$  (d) Syn-2: at $2\Delta t$  (e) Syn-2: at $4\Delta t$

(f) Syn-2: at $10\Delta t$  (g) Syn-3: at $10\Delta t$  (h) Syn-3: at $30\Delta t$  (i) Syn-3: at $50\Delta t$

Figure 2: Comparison of generated data(blue) and ground truth(red) of Synthetic-1((a) to (c)), Synthetic-2((d) to (f)) and Synthetic-3((g) to (i)). In each case, it finally converges to a stationary distribution.

distributions of $x_{50}$ and $x_{500}$. In Synthetic-2, the data is following a complex nonlinear dynamic. We let $\Delta t = 0.01$, $\sigma = \sigma_1 = \sigma_2 = 1$, $\mu_1 = [15, 15]^T$ and $\mu_2 = [-15, -15]^T$. We utilize true $x_0$, $x_3$ and $x_6$ in training process and predict $x_2$, $x_4$ and $x_{10}$. In Synthetic-3, $\Delta t = 0.01$, $\sigma = 1$, we utilize true $x_3$, $x_7$ and $x_{20}$ in training process then predict the distributions of $x_{10}$, $x_{30}$ and $x_{50}$. We also consider cases in higher dimensions: D = 6 and 10. In each high dimensional case, to be convenience, we set every two dimensions follow the dynamics of low dimensional case(D = 2) in each data set. Notice that in Syn-2 and Syn-3, $\hat{x}_t^i$ represents the i-th dimension of $\hat{x}_t$.

**Results:** We first show the capability of our model for learning hidden dynamics of low-dimensional (d = 2) data. As visualized in Figure 2, the generated data(blue) covers all areas of ground truth(red), which demonstrates that our model is able to precisely learn the dynamics and correctly predict the future distribution of data. The samples we predict converge to stationary distributions finally (as the ground truths suggest). We then evaluate three models using Wasserstein distance as error metric for both low-dimensional (d = 2) and high-dimensional (d = 6, 10) data. As reported in Appendix Table 1, our model achieves lower Wasserstein error than the two baseline models in all cases.

### 3.2    Realistic Data – RNA Sequence of Single Cell

In this section, we evaluate our model on a realistic biology data set called Single-cell RNA-seq(Klein et al., 2015), which is typically used for learning the evolvement of cell differentiation. The cell population begins to differentiate at day 0 (D0). Single-cell RNA-seq observations are then sampled at day 0 (D0), day 2 (D2), day 4 (D4) and day 7 (D7). At each time point, the expression of 24,175 genes of several hundreds cells are measured (933, 303, 683 and 798 cells on D0, D2, D4 and D7 respectively). Notice that there is only whole group's distribution but no trajectory information of each gene on different days. We pick 10 gene markers out of 24,175 to make a 10 dimensional data set. In first task we treat gene expression at D0, D4 and D7 as training data to learn the hidden dynamic and predict the distribution of gene expression at D2. In second task we train the model with gene expression at D0, D2 and D4, then predict the distribution of gene expression at D7. We plot the prediction results of two out of ten markers, i.e. Mt1 and Mt2 in Figure 3.

**Experiment Setup:** We set both $f$ and $g$ as fully connected three-hidden-layers neural networks, each layer has 64 nodes. The only activation function we choose is Tanh. The other setups of neural networks and training process are the same with the ones we use in Synthetic data. Notice that in realistic cases, $\Delta t$ and $T/\Delta t$ become hyperparameters, here we choose $\Delta t = 0.05$, $T/\Delta t = 35$, which means the data evolves $10\Delta t$ from D0 to D2 , then $10\Delta t$ from D2 to D4 and finally $15\Delta t$ from D4 to D7. For preprocessing, we apply standard normalization procedures (Hicks et al., 2015) to correct batch effects and use non-negative matrix factorization to impute missing expression levels(Hashimoto et al., 2016; Wang et al., 2018).

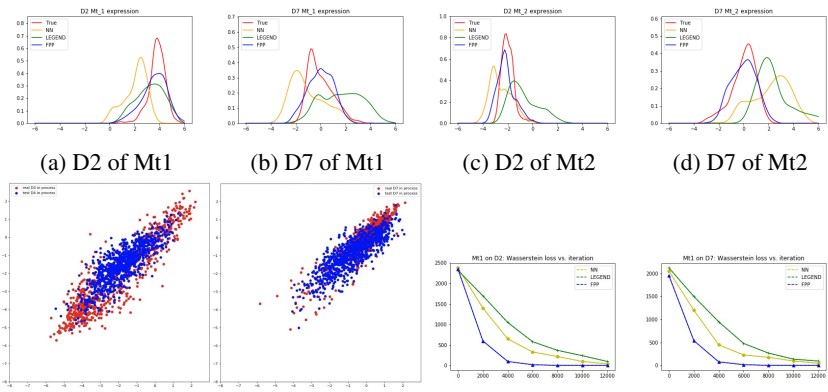

(a) D2 of Mt1    (b) D7 of Mt1    (c) D2 of Mt2    (d) D7 of Mt2

(e) Correlation on D2 (f) Correlation on D7 (g) W-loss of Mt1, D2(h) W-loss of Mt1, D7

Figure 3: (a) and (b): Wasserstein loss as iteration increases of Mt1. (c) to (f): The performance comparision among different models on D2 and D7 of Mt1 and Mt2. (g) and (h): True (red) and predicted (blue) correlations between Mt1(x-axis) and Mt2(y-axis) on D2 (left) and D7 (right).

**Results:** As shown in Table 1 in Appendix, when compared to other baselines, our model achieves lower Wasserstein error on both Mt1 and Mt2 data, which proves that our model is capable of learning the hidden dynamics of the two studied gene expressions. In Figure 3 (a) to (d), we visualized the predicted distributions of the two genes. The distributions of Mt1 and Mt2 predicted by our model (curves in blue) are closer to the true distributions (curves in red) on both D2 and D7. Furthermore, our model precisely indicates the correlations between Mt1 and Mt2, as shown in Figure 3 (e) and (f), which also demonstrates the effectiveness of our model since closer to the true correlation represents better performance. In Figure 3 (g) and (h), we see that with simpler structure, the training process of our model is easier with least computation time.

### 3.3 Realistic Data – Daily Trading Volume

In this section we would like to demonstrate the performance of our model in financial area. Trading volume is the total quantity of shares or contracts traded for specified securities such as stocks, bonds, options contracts, future contracts and all types of commodities. It can be measured on any type of security traded during a trading day or a specified time period. In our case, daily volume of trade is measured on stocks. Predicting traded volume is an essential component in financial research since the traded volume, as a basic component or input of other financial algorithms, tells investors the market's activity and liquidity. The data set we use is the historical traded volume of the stock "JPM". The data covers period from January 2018 to January 2020 and is obtained from Bloomberg. Each day from 14:30 to 20:55, we have 1 observation every 5 minutes, totally 78 observations everyday. Our task is described as follows: given first two years data, we use the traded volume at 14:30, 14:40, 15:05, 15:20 and 16:20 as training data, namely, $x_0, x_2, x_7, x_{10}, x_{22}$ to train our model, then for next 100 days we predict traded volume at 14:35, 15:15, 15:35 and 16:15, namely, $x_1, x_9, x_{13}, x_{21}$. One of baselines we choose is classical rolling means(RM) method, which predicts intraday volume of a particular time interval by the average volume traded in the same interval over the past days. The other one baseline is a kalman filter based model (Chen et al., 2016) that outperforms all available models in predicting intrady trading volume.

**Experiment Setup:** Following similar setup as we did for RNA data set, we utilize the same structures for neural networks here. For hyperparameters we set $\Delta t = 0.02$, $T/\Delta t = 22$, which means it takes one single $\Delta t$ from $x_t$ to $x_{t+1}$. For preprocessing, we rescale data by taking natural logarithm of trading volume, which is a common way in trading volume research. We conduct experiments on two groups to show advantages of our method, for first group we train our model on complete data set, in this case the data has full trajectory; for second group we manually delete some trajectories of the data, for instance, we randomly kick out some samples of $x_0, x_2, x_7, x_{10}, x_{22}$ then follow the same procedures of training and prediction.

**Results:** We present prediction results in Figure 4. As shown in first four figures, with full trajectory, prediction made by RM is almost a straight line, the prediction value bouncing up and down within a

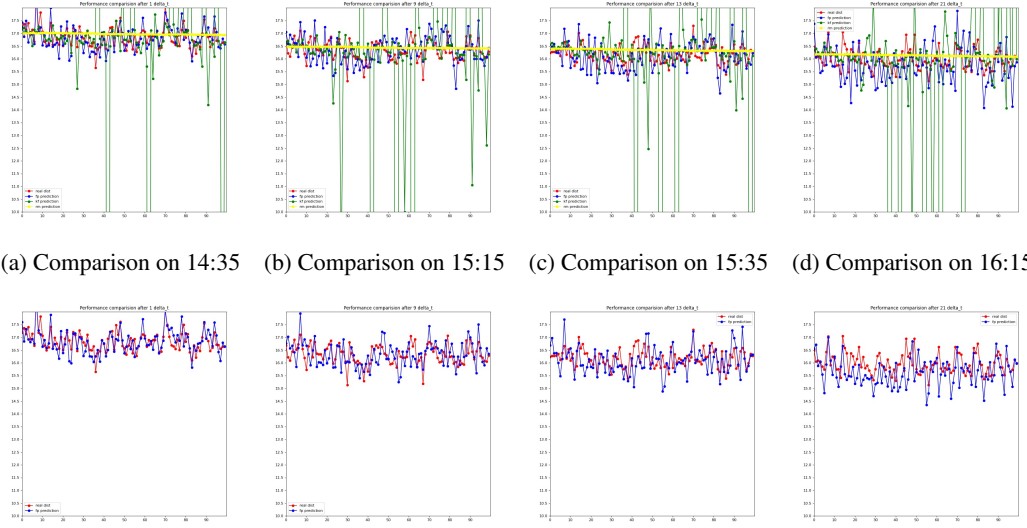

(a) Comparison on 14:35    (b) Comparison on 15:15    (c) Comparison on 15:35    (d) Comparison on 16:15

(e) Comparison on 14:35    (f) Comparison on 15:15    (g) Comparison on 15:35    (h) Comparison on 16:15

Figure 4: (a) to (d): Group A: with full trajectory of training data, predictions of traded volume in next 100 days, RM(yellow) fails to capture the regularities of traded volume in time series, kalman filter based model(green) fails to capture noise information and make reasonable predictions, our model(blue) is able to seize the movements of traded volume and yield better predictions. (e) to (h): Group B: predictions of our model without full trajectory.

very small range, thus this model cannot capture the volume movements, namely, regularities existing in the time series; prediction made by the Kalman filter based model captures the regularities better than RM model, but it fails to deal with noise component existing in the time series, thus some predictions are out of a reasonable range. Traded volume predicted by our model is closer to the real case, moreover, our model captures regularities meanwhile gives stable predictions. Furthermore, without full trajectory, Kalman filter based model fails to be applied here and RM model still fails to capture the regularities, we randomly drop half of the training samples and display predictions made by our model in last four figures of Figure 4, we see our model still works well.

## 4 DISCUSSIONS

In this section we discuss the limitations and extension of our model.

**An essential challenge for recovering the drift term:** Mathematically it is impossible to recover the exact drift term of an SDE if we are only given the information of density evolution on certain time intervals, because there might be infinitely many drift functions to induce the same density evolution. More precisely, suppose $p(x, t)$ solves FPE (equation 1), consider the following equation

$$0 = -\sum_{i=1}^{D} \frac{\partial}{\partial x_i}(u^i(x, t)p(x, t)) + \frac{\sigma^2}{2} \sum_{i=1}^{D} \frac{\partial^2}{\partial x_i^2} p(x, t) \tag{8}$$

One can prove, under mild assumptions, that there are infinitely many vector fields $u(x, t) = (u^1(x, t), ..., u^D(x, t))$ solving equation (8). One can check that the solution to FPE (1) with drift term $g(x, t) + u(x, t)$ is still $p(x, t)$, i.e. the vector field $u(x, t)$ will never affect the density evolution of the dynamic. This illustrates that given the density evolution $p(\cdot, t)$, the solution for drift term is not necessarily unique. This is clearly an essential difficulty of determining the exact drift term from the density. In this study, the main goal is to recover the entire density evolution (i.e. interpolate the density between observation time points) and predict how will the density evolve in the future. As a result, although we cannot always acquire the exact drift term of the dynamic, we can still accurately recover and predict the density evolution. This is still meaningful and may find its application in various scientific domains.

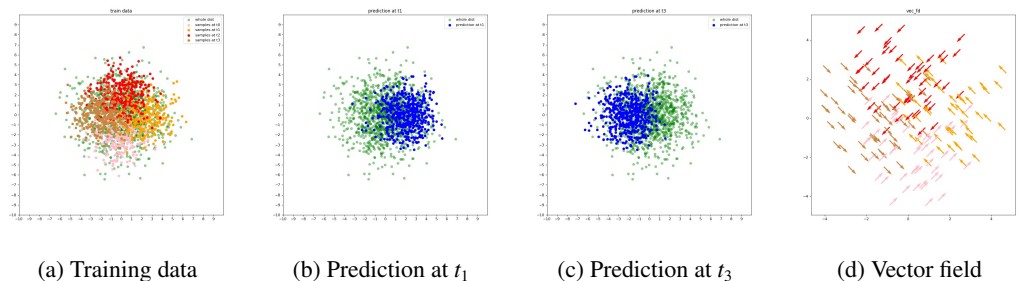

|   |   |   |   |
|---|---|---|---|
| (a) Training data | (b) Prediction at $t_1$ | (c) Prediction at $t_3$ | (d) Vector field |

Figure 5: Results of learning curl field:(a): the whole distribution is indicated by green, from which we choose subsets for training purpose, training samples at $t_0, t_1, t_2$ and $t_3$ are indicated by pink, orange, red and brown respectively. (b) and (c):the prediction at $t_1$ and $t_3$ are indicated by blue points. (d): The vector field generated by our model.

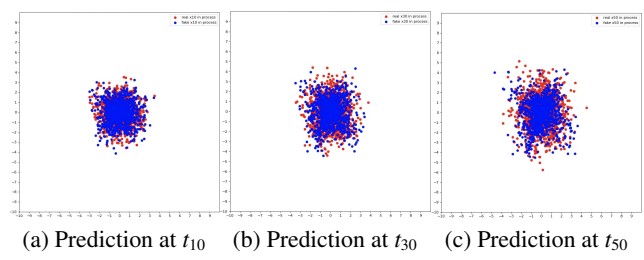

| (a) Prediction at $t_{10}$ | (b) Prediction at $t_{30}$ | (c) Prediction at $t_{50}$ |
|---|---|---|

Figure 6: Results of learning diffusion coefficient.

**Curl Field:** The drift function we showed in the numerical experiments will apparently cause the evolution of the distribution. If the drift function is a curl, namely $g = \nabla \times F$, then the distribution does not change, is it still possible to learn the density evolution? The answer is yes. To demonstrate this point of view, we simulate a curl field $(y, -x)$ induced by $A = [(0, 1), (-1, 0)]$ on a Gaussian distribution (indicated by green points in Figure 5 (a)), here we set noise part as 0. Realizing that our learning is based on samples but not on the density, suppose we observe samples at four time points which does not perfectly follow the distribution (indicated by four colors: pink($t_0$), orange($t_1$), red($t_2$) and brown($t_3$) in Figure 5 (a)), we learn and predict the distribution at $t_1$ and $t_3$ (indicated by blue points in Figure 5 (b) and (c)). We also show the vector field learned by our model in Figure 5 (d). We see that the predictions and vector field all satisfy the correct curl field pattern.

**Learning Diffusion Coefficient:** Our framework also works for learning unknown diffusion coefficient in the Itô process. As an extension of our work, if we approximate the diffusion coefficient with a neural network $\sigma_\eta$ (with parameters $\eta$), we revise the operator $\mathcal{F}$ as:

$$\mathcal{F}_f(X) = \frac{1}{N} \sum_{k=1}^{N} \left( \sum_{i=1}^{D} g_\omega^i(\boldsymbol{x}^{(k)}) \frac{\partial}{\partial x_i} f(\boldsymbol{x}^{(k)}) + \sum_{i=1}^{D} \left( \sum_{j=1}^{D} \frac{1}{2} (\sigma_\eta^{ij}(\boldsymbol{x}^{(k)}))^2 \right) \frac{\partial^2}{\partial x_i^2} f(\boldsymbol{x}^{(k)}) \right) \tag{9}$$

which can be derived by the same technique we used to derive Proposition 1. We test this formulation on a synthetic data set, where we only consider diffusion influence, namely, drift term in Equation (8) is ignored. We set the ground truth of diffusion coefficient as $\sigma = [(1, 0), (0, 2)]$. We design the neural network as a simple one fully connected layer with 32 nodes, then show our result in Figure 6, we see that the predictions(blue) follow the same pattern as the ground truth(red) does.

## 5 CONCLUSION

In this paper, we formulate a novel method to recover the hidden dynamics from aggregate data. In particular, our work shows one can simulate the evolving process of aggregate data as an Itô process, in order to investigate aggregate data, we derive a new model that employs the weak form of FPE as well as the framework of WGAN. Furthermore, in Appendix we prove the theoretical guarantees of the error bound of our model. Finally we demonstrate our model through experiments on three synthetic data sets and two real-world data sets.

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

## A  APPENDIX

Table 1: The Wasserstein error of different models on Synthetic-1/2/3 and RNA-sequence data sets.

| Data | Task | Dimension | NN | LEGEND | Ours |
|---|---|---|---|---|---|
| Syn–1 | $x_{50}$ | 2 | 1.37 | 0.44 | **0.05** |
| | | 6 | 4.79 | 2.32 | **0.06** |
| | | 10 | 9.13 | 2.89 | **0.10** |
| | $x_{500}$ | 2 | 0.84 | 0.18 | **0.03** |
| | | 6 | 3.28 | 0.30 | **0.03** |
| | | 10 | 8.05 | 1.79 | **0.09** |
| Syn–2 | $x_4$ | 2 | 5.53 | 1.27 | **0.05** |
| | | 6 | 9.97 | 4.20 | **0.08** |
| | | 10 | 16.51 | 6.08 | **0.12** |
| | $x_{10}$ | 2 | 1.68 | 0.53 | **0.05** |
| | | 6 | 4.84 | 1.08 | **0.07** |
| | | 10 | 9.23 | 2.85 | **0.11** |
| Syn–3 | $x_{30}$ | 2 | 4.13 | 1.29 | **0.08** |
| | | 6 | 6.40 | 3.16 | **0.17** |
| | | 10 | 11.76 | 8.53 | **0.25** |
| | $x_{50}$ | 2 | 3.05 | 0.87 | **0.12** |
| | | 6 | 6.72 | 1.52 | **0.16** |
| | | 10 | 9.81 | 3.55 | **0.23** |
| RNA-Mt1 | D2 | 10 | 33.86 | 10.28 | **4.23** |
| | D7 | 10 | 12.69 | 7.21 | **2.92** |
| RNA-Mt2 | D2 | 10 | 31.45 | 13.32 | **4.04** |
| | D7 | 10 | 11.58 | 7.89 | **1.50** |

## B  ERROR ANALYSIS

In this section, we provide an error analysis of our model. Suppose the hidden dynamics is driven by $g_r(x)$, the dynamics that we learn from data is $g_f(x)$, then original Itô process, Euler processes computed by true $g_r$ and estimated $g_f$ are:

$$dX = g(X)dt + \sigma dW$$

$$x^r_{t+\Delta t} = x^r_t + g_r(x^r_t)\Delta t + \sigma \sqrt{\Delta t}\mathcal{N}(0,1)$$

$$x^f_{t+\Delta t} = x^f_t + g_f(x^f_t)\Delta t + \sigma \sqrt{\Delta t}\mathcal{N}(0,1)$$

where $X$ is the ground truth, $x^r$ is computed by true $g_r$ and $x^f$ is computed by estimated $g_f$. Estimating the error between original Itô process and its Euler form can be very complex, hence we cite the conclusion from (Milstein & Tretyakov, 2013) and focus more on the error between original form and our model.

**Lemma 2.** *With the same initial $X_{t_0} = x_{t_0} = x_0$, if there is a global Lipschitz constant $K$ which satisfies:*

$$|g(x,t) - g(y,t)| \leq K|x - y|$$

*then after n steps, the expectation error between Itô process $x_{t_n}$ and Euler forward process $x^r_{t_n}$ is:*

$$\mathbb{E}|x_{t_n} - x^r_{t_n}| \leq K\left(1 + \mathbb{E}|X_0|^2\right)^{1/2}\Delta t$$

Lemma 2 illustrates that the expectation error between original Itô process and its Euler form is not related to total steps $n$ but time step $\Delta t$.

**Proposition 3.** *With the same initial $x_0$, suppose the generalization error of neural network $g$ is $\varepsilon$ and existence of global Lipschitz constant $K$:*

$$|g(\boldsymbol{x}) - g(\boldsymbol{y})| \leq K|\boldsymbol{x} - \boldsymbol{y}|$$

*then after $n$ steps with step size $\Delta t = T/n$, the expectation error between Itô process $\boldsymbol{x}_{t_n}$ and approximated forward process $\boldsymbol{x}_{t_n}^f$ is bounded by:*

$$\mathbb{E}|\boldsymbol{x}_{t_n} - \boldsymbol{x}_{t_n}^f| \leq \frac{\varepsilon}{K}(e^{KT} - 1) + K(1 + \mathbb{E}|\boldsymbol{x}_0|^2)^{1/2}\Delta t \tag{10}$$

Proposition 3 implies that besides time step size $\Delta t$, our expectation error interacts with three factors, generalization error, Lipschitz constant of $g$ and total time length. In our experiments, we find the best way to decrease the expectation error is reducing the value of $K$ and $n$.

## C    PROOFS

### C.1    PROOF OF PROPOSITION 1

*Proof.* Suppose $\hat{x}_{t_m}^{(k)}$ and $\hat{x}_{t_{m-1}}^{(k)}$ are our observed samples at $t_m$ and $t_{m-1}$ respectively, then expectations could be approximated by:

$$\mathbb{E}_{\boldsymbol{x} \sim \hat{p}(\boldsymbol{x}, t_m)}[f(\boldsymbol{x})] = \int f(\boldsymbol{x})\hat{p}(\boldsymbol{x}, t_m)d\boldsymbol{x} \approx \frac{1}{N}\sum_{k=1}^{N} f(\hat{x}_{t_m}^{(k)}) \tag{11}$$

$$
\begin{aligned}
\mathbb{E}_{\boldsymbol{x} \sim \tilde{p}(\boldsymbol{x}, t_m)}[f(\boldsymbol{x})] &= \int f(\boldsymbol{x})\tilde{p}(\boldsymbol{x}, t_m)d\boldsymbol{x} = \int f(\boldsymbol{x})\left[\hat{p}(\boldsymbol{x}, t_{m-1}) + \int_{t_{m-1}}^{t_m} \frac{\partial p(\boldsymbol{x}, \tau)}{\partial t}d\tau\right]d\boldsymbol{x} \\
&= \int f(\boldsymbol{x})\hat{p}(\boldsymbol{x}, t_{m-1})d\boldsymbol{x} + \int f(\boldsymbol{x})\int_{t_{m-1}}^{t_m} \frac{\partial p(\boldsymbol{x}, \tau)}{\partial t}d\tau d\boldsymbol{x} \\
&\approx \frac{1}{N}\sum_{k=1}^{N} f(\hat{x}_{t_{m-1}}^{(k)}) + \underbrace{\int f(\boldsymbol{x})\int_{t_{m-1}}^{t_m}\left\{-\sum_{i=1}^{D}\frac{\partial}{\partial x_i}\left[g_\omega^i(\boldsymbol{x})p(\boldsymbol{x}, \tau)\right] + \frac{1}{2}\sigma^2\sum_{i=1}^{D}\frac{\partial^2}{\partial x_i^2}p(\boldsymbol{x}, \tau)\right\}d\tau d\boldsymbol{x}}_{I}
\end{aligned}
\tag{12}
$$

Then for the second term $I$ above, it is difficult to calculate directly, but we can use integration by parts to rewrite $I$ as:

$$
\begin{aligned}
I &= \int_{t_{m-1}}^{t_m}\int\left[\sum_{i=1}^{D}-f(\boldsymbol{x})\frac{\partial}{\partial x_i}g_\omega^i(\boldsymbol{x})p(\boldsymbol{x}, \tau) + \frac{1}{2}\sigma^2\sum_{i=1}^{D}f(\boldsymbol{x})\frac{\partial^2}{\partial x_i^2}p(\boldsymbol{x}, \tau)\right]d\boldsymbol{x}d\tau \\
&= \int_{t_{m-1}}^{t_m}\int\left[\sum_{i=1}^{D}g_\omega^i(\boldsymbol{x})p(\boldsymbol{x}, \tau)\frac{\partial}{\partial x_i}f(\boldsymbol{x}) + \frac{1}{2}\sigma^2\sum_{i=1}^{D}p(\boldsymbol{x}, \tau)\frac{\partial^2}{\partial x_i^2}f(\boldsymbol{x})\right]d\boldsymbol{x}d\tau \\
&= \int_{t_{m-1}}^{t_m}\left(\mathbb{E}_{\boldsymbol{x} \sim p(\boldsymbol{x}, \tau)}\left[\sum_{i=1}^{D}g_\omega^i(\boldsymbol{x})\frac{\partial}{\partial x_i}f(\boldsymbol{x})\right] + \mathbb{E}_{\boldsymbol{x} \sim p(\boldsymbol{x}, \tau)}\left[\frac{1}{2}\sigma^2\sum_{i=1}^{D}\frac{\partial^2}{\partial x_i^2}f(\boldsymbol{x})\right]\right)d\tau \\
&\approx \int_{t_{m-1}}^{t_m}\frac{1}{N}\sum_{k=1}^{N}\left(\sum_{i=1}^{D}g_\omega^i(x^{(k)})\frac{\partial}{\partial x_i}f(x^{(k)}) + \frac{1}{2}\sigma^2\sum_{i=1}^{D}\frac{\partial^2}{\partial x_i^2}f(x^{(k)})\right)d\tau
\end{aligned}
\tag{13}
$$

To approximate the integral from $t_{m-1}$ to $t_m$, we adopt trapezoid rule, then we could rewrite the expectation in Equation (12) as:

$$\mathbb{E}_{\boldsymbol{x} \sim \tilde{p}(\boldsymbol{x}, t_m)}[f(\boldsymbol{x})] \approx \frac{1}{N} \sum_{k=1}^{N} f(\hat{x}_{t_{m-1}}^{(k)}) + \frac{\Delta t}{2} \left[ \frac{1}{N} \sum_{k=1}^{N} \left( \sum_{i=1}^{D} g_\omega^i(\hat{x}_{t_{m-1}}^{(k)}) \frac{\partial}{\partial x_i} f(\hat{x}_{t_{m-1}}^{(k)}) + \frac{1}{2} \sigma^2 \sum_{i=1}^{D} \frac{\partial^2}{\partial x_i^2} f(\hat{x}_{t_{m-1}}^{(k)}) \right) \right.$$

$$+ \frac{1}{N} \sum_{k=1}^{N} \left( \sum_{i=1}^{D} g_\omega^i(\tilde{x}_{t_m}^{(k)}) \frac{\partial}{\partial x} f(\tilde{x}_{t_m}^{(k)}) + \frac{1}{2} \sigma^2 \sum_{i=1}^{D} \frac{\partial^2}{\partial x_i^2} f(\tilde{x}_{t_m}^{(k)}) \right) \Bigg]$$

$$= \frac{1}{N} \sum_{k=1}^{N} f(\hat{x}_{t_{m-1}}^{(k)}) + \frac{\Delta t}{2} \left[ \mathcal{F}_f(\hat{X}_{m-1}) + \mathcal{F}_f(\tilde{X}_m) \right] \tag{14}$$

We subtract (11) by (14) to finish the proof. □

## C.2 Proof of Proposition 2

*Proof.* Given initial $\hat{\boldsymbol{x}}_{t_0}$, we generate $\tilde{\boldsymbol{x}}_{t_1}$, $\tilde{\boldsymbol{x}}_{t_2}$, $\tilde{\boldsymbol{x}}_{t_3}$ ... $\tilde{\boldsymbol{x}}_{t_n}$ sequentially by Equation (6). Then the expectations can be rewritten as:

$$\mathbb{E}_{\boldsymbol{x} \sim \hat{p}(\boldsymbol{x}, t_n)}[f(\boldsymbol{x})] = \int f(\boldsymbol{x}) \hat{p}(\boldsymbol{x}, t_n) d\boldsymbol{x} \approx \frac{1}{N} \sum_{k=1}^{N} f(\hat{x}_{t_n}^{(k)}) \tag{15}$$

$$\mathbb{E}_{\boldsymbol{x} \sim \tilde{p}(\boldsymbol{x}, t_n)}[f(\boldsymbol{x})] \approx \frac{1}{N} \sum_{k=1}^{N} f(\hat{x}_{t_0}^{(k)}) + \int_{t_0}^{t_1} \frac{1}{N} \sum_{k=1}^{N} \left[ \sum_{i=1}^{D} g_\omega^i(x^{(k)}) \frac{\partial}{\partial x_i} f(x^{(k)}) + \frac{1}{2} \sigma^2 \sum_{i=1}^{D} \frac{\partial^2}{\partial x_i \partial x_j} f(x^{(k)}) \right] d\tau$$

$$+ \int_{t_1}^{t_2} \frac{1}{N} \sum_{k=1}^{N} \left[ \sum_{i=1}^{D} g_\omega^i(x^{(k)}) \frac{\partial}{\partial x_i} f(x^{(k)}) + \frac{1}{2} \sigma^2 \sum_{i=1}^{D} \frac{\partial^2}{\partial x_i^2} f(x^{(k)}) \right] d\tau + ...$$

$$+ \int_{t_{n-1}}^{t_n} \frac{1}{N} \sum_{k=1}^{N} \left[ \sum_{i=1}^{n} g_\omega^i(x^{(k)}) \frac{\partial}{\partial x_i} f(x^{(k)}) + \frac{1}{2} \sigma^2 \sum_{i=1}^{n} \frac{\partial^2}{\partial x_i^2} f(x^{(k)}) \right] d\tau \tag{16}$$

which is:

$$\mathbb{E}_{\boldsymbol{x} \sim \tilde{p}(\boldsymbol{x}, t_n)}[f(\boldsymbol{x})] \approx \frac{1}{N} \sum_{k=1}^{N} f(\hat{x}_{t_0}^{(k)}) + \frac{\Delta t}{2} \left[ \mathcal{F}_f(\hat{X}_0) + \mathcal{F}_f(\tilde{X}_1) \right] + \frac{\Delta t}{2} \left[ \mathcal{F}_f(\tilde{X}_1) + \mathcal{F}_f(\tilde{X}_2) \right] + ...$$

$$+ \frac{\Delta t}{2} \left[ \mathcal{F}_f(\tilde{X}_{n-1}) + \mathcal{F}_f(\tilde{X}_n) \right] \tag{17}$$

Finally it comes to:

$$\mathbb{E}_{\boldsymbol{x} \sim \tilde{p}(\boldsymbol{x}, t_n)}[f(\boldsymbol{x})] \approx \frac{1}{N} \sum_{k=1}^{N} f(\hat{x}_{t_0}^{(k)}) + \frac{\Delta t}{2} \left( \mathcal{F}_f(\hat{X}_0) + \mathcal{F}_f(\tilde{X}_n) + 2 \sum_{s=1}^{n-1} \mathcal{F}_f(\tilde{X}_s) \right) \tag{18}$$

We subtract (15) by (18) to finish the proof. □

## C.3 Proof of Error Analysis

*Proof.* The proof process of Lemma 2 is quite long and out of the scope of this paper, for more details please see first two chapters in reference book (Milstein & Tretyakov, 2013). While for the proof of Proposition 3, with initial $X$ and first one-step iteration:

$$\begin{cases} \boldsymbol{x}_{t_0}^r = \boldsymbol{x}_{t_0} \\ \boldsymbol{x}_{t_0}^f = \boldsymbol{x}_{t_0} \end{cases} \tag{19}$$

$$\begin{cases} \boldsymbol{x}_{t_1}^r = \boldsymbol{x}_{t_0}^r + g_r(\boldsymbol{x}_{t_0}^r)\Delta t + \sigma \sqrt{\Delta t} \mathcal{N}(0,1) \\ \boldsymbol{x}_{t_1}^f = \boldsymbol{x}_{t_0}^f + g_f(\boldsymbol{x}_{t_0}^f)\Delta t + \sigma \sqrt{\Delta t} \mathcal{N}(0,1) \end{cases} \tag{20}$$

Then we have:

$$\mathbb{E}|\boldsymbol{x}_{t_0}^r - \boldsymbol{x}_{t_0}^f| = \mathbb{E}|\boldsymbol{x}_{t_0} - \boldsymbol{x}_{t_0}| = 0 \tag{21}$$

$$
\begin{aligned}
\mathbb{E}|\boldsymbol{x}_{t_1}^r - \boldsymbol{x}_{t_1}^f| &= \mathbb{E}|\boldsymbol{x}_{t_0}^r - \boldsymbol{x}_{t_0}^f + g_r(\boldsymbol{x}_{t_0}^r)\Delta t - g_f(\boldsymbol{x}_{t_0}^f)\Delta t + \sigma\sqrt{\Delta t}\mathcal{N}(0,1) - \sigma\sqrt{\Delta t}\mathcal{N}(0,1)| \\
&\leq \mathbb{E}|\boldsymbol{x}_{t_0}^r - \boldsymbol{x}_{t_0}^f| + \mathbb{E}|g_r(\boldsymbol{x}_{t_0}^r) - g_f(\boldsymbol{x}_{t_0}^f)|\Delta t \\
&= \mathbb{E}|g_r(\boldsymbol{x}_{t_0}^r) - g_f(\boldsymbol{x}_{t_0}^r) + g_f(\boldsymbol{x}_{t_0}^r) - g_f(\boldsymbol{x}_{t_0}^f)|\Delta t \\
&\leq \mathbb{E}|g_r(\boldsymbol{x}_{t_0}^r) - g_f(\boldsymbol{x}_{t_0}^r)|\Delta t + \mathbb{E}|g_f(\boldsymbol{x}_{t_0}^r) - g_f(\boldsymbol{x}_{t_0}^f)|\Delta t \\
&\leq \varepsilon\Delta t + \mathbb{E}|g_f(\boldsymbol{x}_{t_0}^r) - g_f(\boldsymbol{x}_{t_0}^f)|\Delta t \\
&= \varepsilon\Delta t + \mathbb{E}|g_f'(\boldsymbol{x}_{t_0}^\xi)(\boldsymbol{x}_{t_0}^r - \boldsymbol{x}_{t_0}^f)|\Delta t \qquad\qquad (\boldsymbol{x}_{t_0}^\xi \in [\boldsymbol{x}_{t_0}^r, \boldsymbol{x}_{t_0}^f]) \\
&\leq \varepsilon\Delta t + K\mathbb{E}|\boldsymbol{x}_{t_0}^r - \boldsymbol{x}_{t_0}^f|\Delta t \\
&= \varepsilon\Delta t
\end{aligned}
\tag{22}
$$

Follow the pattern we have:

$$
\begin{cases}
\boldsymbol{x}_{t_2}^r = \boldsymbol{x}_{t_1}^r + g_r(\boldsymbol{x}_{t_1}^r)\Delta t + \sigma\sqrt{\Delta t}\mathcal{N}(0,1) \\
\boldsymbol{x}_{t_2}^f = \boldsymbol{x}_{t_1}^f + g_f(\boldsymbol{x}_{t_1}^f)\Delta t + \sigma\sqrt{\Delta t}\mathcal{N}(0,1)
\end{cases}
\tag{23}
$$

$$\dots$$

$$
\begin{cases}
\boldsymbol{x}_{t_n}^r = \boldsymbol{x}_{t_{n-1}}^r + g_r(\boldsymbol{x}_{t_{n-1}}^r)\Delta t + \sigma\sqrt{\Delta t}\mathcal{N}(0,1) \\
\boldsymbol{x}_{t_n}^f = \boldsymbol{x}_{t_{n-1}}^f + g_f(\boldsymbol{x}_{t_{n-1}}^f)\Delta t + \sigma\sqrt{\Delta t}\mathcal{N}(0,1)
\end{cases}
\tag{24}
$$

Which leads to:

$$
\begin{aligned}
\mathbb{E}|\boldsymbol{x}_{t_2}^r - \boldsymbol{x}_{t_2}^f| &= \mathbb{E}|\boldsymbol{x}_{t_1}^r - \boldsymbol{x}_{t_1}^f + g_r(\boldsymbol{x}_{t_1}^r)\Delta t - g_f(\boldsymbol{x}_{t_1}^f)\Delta t + \sigma\sqrt{\Delta t}\mathcal{N}(0,1) - \sigma\sqrt{\Delta t}\mathcal{N}(0,1)| \\
&\leq \mathbb{E}|\boldsymbol{x}_{t_1}^r - \boldsymbol{x}_{t_1}^f| + \mathbb{E}|g_r(\boldsymbol{x}_{t_1}^r) - g_f(\boldsymbol{x}_{t_1}^f)|\Delta t \\
&\leq \mathbb{E}|\boldsymbol{x}_{t_1}^r - \boldsymbol{x}_{t_1}^f| + \varepsilon\Delta t + K\mathbb{E}|\boldsymbol{x}_{t_1}^r - \boldsymbol{x}_{t_1}^f|\Delta t \\
&\leq (1 + K\Delta t)\varepsilon\Delta t + \varepsilon\Delta t
\end{aligned}
\tag{25}
$$

$$\dots$$

$$\mathbb{E}|\boldsymbol{x}_{t_n}^r - \boldsymbol{x}_{t_n}^f| \leq \varepsilon\Delta t \sum_{i=0}^{n-1}(1 + K\Delta t)^i \tag{26}$$

Now let $S = \sum_{i=0}^{n-1}(1 + K\Delta t)^i$, then consider followings:

$$
\begin{aligned}
S(K\Delta t) &= S(1 + K\Delta t) - S \\
&= \sum_{i=1}^{n}(1 + K\Delta t)^i - \sum_{i=0}^{n-1}(1 + K\Delta t)^i \\
&= (1 + K\Delta t)^n - 1 \\
&= (1 + K\frac{T}{n})^n - 1 \\
&\leq e^{KT} - 1
\end{aligned}
\tag{27}
$$

Finally we have:

$$\mathbb{E}|\boldsymbol{x}_{t_n}^r - \boldsymbol{x}_{t_n}^f| \leq \frac{\varepsilon}{K}(e^{KT} - 1) \tag{28}$$

$$\mathbb{E}|\boldsymbol{x}_{t_n} - \boldsymbol{x}_{t_n}^f| \leq \frac{\varepsilon}{K}(e^{KT} - 1) + K(1 + E|\boldsymbol{x}_0|^2)^{1/2}\Delta t \tag{29}$$

$\square$

