# OpenReview forum: "Learning Stochastic Behaviour from Aggregate Data"
_ICLR.cc/2021/Conference — Reject_

### Official Review · AnonReviewer1 · 2020-10-23
**A paper on an interesting topic, but with strange similarities with an already-published article**

**Rating:** 4
**Confidence:** 4

**Review:**

The paper is about an interesting setting, where observations correspond to lots of individuals evolving over time. The twist is that individual identifiers are not available.  Thus from one time point to the next, the observer does not know which individual is which. The problem is to learn a time series model from this, and models considered here are in the form of an SDE.

In my first reading of the paper, I thought the topic was very interesting but I struggled to follow some parts of the manuscript, which I thought was overly unclear. Some of the notation is undefined or ill-defined, the general reasoning is hard to follow, the notation does not help identifying the objects that are observed from the objects that are simulated, there is a lack of acknowledgement of limitations of the proposed method, the synthetic examples are not well-motivated, and various other fairly standard concerns.

However, during my second reading I looked at the reference Wang et al, "Learning Deep Hidden Nonlinear Dynamics from Aggregate Data", published in UAI in 2018.  From this I got more serious concerns.

The authors' abstract starts with "Learning nonlinear dynamics from aggregate data is a challenging problem since the full trajectory of each individual is not available, namely, the individual observed at one time point may not be observed at next time point, or the identity of individual is unavailable due to technical limitations, experimental cost and/or privacy issues."

Wang et al's abstract mentions: "However, in most of the practical applications, these requirements are unrealistic: the evolving dynamics may be too complex to be modeled directly on observations, and individual-level trajectories may not be available due to technical limitations, experimental costs and/or privacy issues."

The authors write "In the work of Hashimoto et al. (2016), a stochastic differential equation(SDE) was adopted to capture the dynamics of particles directly on observations, in particular the drift coefficient is parameterized by a recurrent neural network."

Wang et al. write: "Modeling the dynamics on aggregate observations have been investigated recently in (Hashimoto et al., 2016), where a stochastic differential equation (SDE) has been used to capture the transition directly on observations Yt."

The authors write: "There are many existing models to learn dynamics of full-trajectory data. Popular ones include Hidden Markov Model (HMM)(Alshamaa et al., 2019; Eddy, 1996), Kalman Filter (KF)(Farahi & Yazdi, 2020; Harvey, 1990; Kalman, 1960) and Particle Filter (PF) (Santos et al., 2019; Djuric et al., 2003). These models and their variants (Deriche et al., 2020; Fang et al., 2019; Hefny et al., 2015; Langford et al., 2009) require full trajectories of each individual, which may not be directly applicable to the aggregate data as we mentioned earlier. "

Wang et al. write: "Existing models such as Hidden Markov Model (HMM) (Eddy, 1996), Kalman Filter (KF) (Harvey, 1990) and Particle Filter (PF) (Djuric et al., 2003) are popular methods with hidden variables. However, these models and their variants (Langford et al., 2009; Hefny et al., 2015) require individual-level trajectories, which may not be available, as was mentioned earlier."

--

Based on these strong similarities I do not think that the proposed manuscript is suitable for publication.

---

> ### Author Response · Authors · 2020-11-23
> **Response to Reviewer1**
>
> Thank you for your comments, we appreciate your time and important suggestions. For your concerns, here are our point-by-point responses.
>
>
>
> 1.The reviewer commented “In my first reading of the paper, I thought the topic was very interesting but I struggled to follow some parts of the manuscript, which I thought was overly unclear. Some of the notation is undefined or ill-defined, the general reasoning is hard to follow, the notation does not help identifying the objects that are observed from the objects that are simulated.”
>
> Response: We rewrote the derivation part (section 2 in the paper) in order to make it clearer. The current reasoning has been revised to be more coherent and rigorous. Hopefully it will be easier to follow. Despite that, we also improved our narrative language and fixed several unclear notations which may cause confusions to the readers.
>
>
>
> 2.The reviewer commented “There is a lack of acknowledgement of limitations of the proposed method.”
>
> Response: This a good suggestion. One obvious limitation is that we can’t recover the trajectory information from the learning. Another limitation is that in theory our method can only learn a drift that may be a curl field different from the real drift term. This is because there may be infinitely many drift functions to induce the same density evolution. In other words, given the density evolution $p(·,t)$, the solution for drift term is not necessarily unique. This is an intrinsic difficulty of determining the exact drift term from the density. To better illustrate this point, we added this limitation in a new section “Discussions”. We also added a new experiment: a Gaussian distribution is driven by a curl field, thus at any time the distribution doesn’t change, in such a case if we only know the information of the distribution, it is impossible to recover the drift function as a curl field. However, if we know enough samples from the Gaussian distribution and these samples doesn’t follow the distribution perfectly, we demonstrate that our algorithm is able to capture the correct vector field.
>
>
>
> 3.The reviewer commented “The synthetic examples are not well-motivated, and various other fairly standard concerns.”
>
> Response: In this work we aim to predict the future distribution of the aggregate data, with the help of Fokker Planck Equation and Wasserstein GAN (WGAN). Our method begins with using the weak form of Fokker Planck Equation to estimate the densities, then we compare generated distribution and real distribution within the WGAN framework. For synthetic data, we want to prove the efficiency of our method in both linear and nonlinear cases, it is shown that our method is able to capture and predict the distribution evolutions.
>
>
>
> 4.The reviewer commented: “However, during my second reading I looked at the reference Wang et al, "Learning Deep Hidden Nonlinear Dynamics from Aggregate Data", published in UAI in 2018. From this I got more serious concerns. … ”
>
> Response: Thank you for pointing out this important aspect. We revised our paper carefully to avoid the similarities in the new version.  Our work and Wang’s work are in the same domain, the question we both want to solve is how to model the hidden dynamic from aggregate data, we all select Wasserstein distance to compare distributions. For that reason, the description of the problem have similarities. However, our strategy and algorithm are totally different. Wang did it from the perspective of Hidden Markov Model, while we do it from the view of the weak form of Fokker Planck Equation.
> .

---

### Official Review · AnonReviewer4 · 2020-10-28
**Recommendation to Accept**

**Rating:** 8
**Confidence:** 3

**Review:**

This paper proposes a new approach to learn the dynamics of density evolution of objects from aggregated data. The basic idea is to derive a closed-form Wasserstein distance between the empirical data distribution and the predicted distribution generated from the weak form of Fokker Planck Equation (FPE). Based on this measure, an objective function is developed to learn the underlying drift coefficients and the discriminator simultaneously using neural networks. Some theoretical results are provided and numerical experiments are carried out to illustrate the effectiveness of the proposed method.

The paper is overall well written and solves an important problem. I have only a few minor questions.

1. How prediction is made based on the estimated model? One step ahead or multiple-step ahead prediction? Please provide more details on this issue.

2. Some notations are unclear in Theorems 1-1. In Theorem 1, what are the definitions of $N$ and $x^{(k)}$? Do you generate data for N times? In Theorem 2, I  assume that $n$ stands for $n$ steps ahead from t_{m_0}?

---

> ### Author Response · Authors · 2020-11-23
> **Response to Reviewer4**
>
> Thank you for your comments, we appreciate your time and constructive suggestions. For your concerns, here are our responses.
>
>
> 1.For training and prediction process, starting with true $x_0$, the data at each future time point is generated and predicted by the Euler-Maruyama scheme of the SDE. For example, if we know true $x_0$, $x_2$, $x_5$,  $x_8$, $x_{10}$, $x_{15}$ and $x_{20}$, during training process we generate $x_5$, $x_{10}$ and $x_{20}$, all starting with $x_0,$ by Euler-Maruyama scheme of the SDE with updated drift function. When the training is done, still start with $x_0$, we predict $x_2$, $x_8$ and $x_{15}$ by Euler-Maruyama scheme of the SDE with the final learned drift function, then we compare them with the ground truth. Relevant contents are addressed in our experiments section.
>
>
> 2.Thank you for pointing out this, here “N” represents the sample size at each time point, “D” represents the dimension of the data, “n” is the number of steps we move forward. We clarify these in Proposition 1 and 2, since we change Theorem 1 and 2 to Proposition 1 and 2. The derivations and notations are also revised to make the paper more readable.

---

### Official Review · AnonReviewer3 · 2020-10-28
**Interesting problem, but requires more work**

**Rating:** 5
**Confidence:** 4

**Review:**

The paper addresses an interesting problem: learning stochastic dynamics from aggregate data. The aggregate data refers to the setting when the data is anonymized. For example, one has data about the location of the birds at different instants in time, but the birds are not labeled and can not be distinguished from each other. For the purpose of learning, the paper proposes to use a generator, in the setting of WGAN, to learn the drift term in the dynamics. The novelty of the proposed approach is to consider the weak form of the Fokker-Planck equation to express the loss function.

The problem is interesting, but I found several issues about the proposed approach that need to be addressed.

1- The paper consider a special setting where the dynamics is given by a diffusion process. It is assumed that the diffusion coefficient is known and the objective is to learn the drift function. I think this is a strong assumption and learning diffusion coefficient is also very important and need to be included in this work.

2- Learning the drift function from aggregate data may have fundamental limitation that the paper needs to address. It seems impossible to learn mixing terms (the terms in the dynamics that do not effect the distribution) from aggregate data. In other words, even in and ideal setting with infinite capacity for neural network and infinite amount of data, the learned drift function differs from the actual one. This can be verified by running a simple experiment with dynamics [[0,1][-1,0]] so that the distribution does not change and the learned drift function should be zero.

3- In the proposed formulation, eq (8), it is not clear which one is real data and which one is generated data.

4- Theorem 1 is not stated correctly and it is missleading. It states that eq (7) is the Wasserstein distance. However, eq (7) is just an approximation. And I really don't think this should be stated as a result in a Theorem. This is basic definition of Wasserstein distance and eq (1) and it is more appropriate to be presented as part of text.  Also, Lemma 1 is well-known and a reference should be given.

5- The numerical experiments are nice, but they should also consider the long-term behavior: If the learned dynamics results to the correct stationary distribution or not. Because the stationary distribution is one of the important characteristics of a stochastic dynamics.

---

> ### Author Response · Authors · 2020-11-23
> **Response to Reviewer3**
>
> Thank you for your comments on our work, we appreciate your time and constructive suggestions. Here are our point-by-point responses to your concerns.
>
>
> 1.Learning diffusion term is of great interest to us as well. We consider it as a natural extension of the work in this paper. We conducted an experiment by treating diffusion coefficient as a neural network, and the results seem to be promising. We have added this part to the new version of the paper.
>
>
> 2.This is a great point, thanks for pointing out for us. Mathematically, it is impossible to recover the exact drift term of an SDE if we are only given the information of density evolution on certain time interval. This is because the solution for drift term is not necessarily unique for a given density evolution $p(·,t)$. There might be infinitely many drift functions that can induce the same density evolution. This is clearly an essential difficulty of determining the exact drift term from the density. In this study, our main goal is to recover the entire density evolution (i.e. interpolate the density between observation time points) and predict the density in the future rather than only learning for the drift function. As a result, although we cannot always acquire the exact drift term of the dynamic, we can still accurately recover and predict the density evolution. To better illustrate our view, following your suggestion, we added an experiment in section “Discussions”. Saying that a Gaussian distribution is driven by a curl field, thus at any time the distribution doesn’t change, in such a case if we only know the information of the distribution, it is impossible to recover the drift function as a curl field. However, if we know enough samples from the Gaussian distribution and these samples doesn’t follow the distribution perfectly, we demonstrated that our algorithm is able to capture the correct vector field.
>
>
> 3.We revised $\hat{x}$ as true data and $\tilde{x}$ as generated data.
>
>
> 4.We changed “Theorems” to “Propositions” since the Wasserstein distance here is an approximation form. Also, we changed the words to “the Wasserstein distance between … can be approximated by…”. We wrote Ito process in the text and the reference of Lemma 1 is added. We also improved the derivations and notations so as to make the paper easier to follow.
>
>
> 5.Sorry for the unclear statement, in all cases of numerical experiments, the distributions we predict finally converge to stationary distributions (as ground truths suggest). Syn-1 and syn-3 both converge to the distributions as their last figure shows, syn-2 converges to the defined mean slowly, we only show the evolution trend of the distribution due to the limitation of the paper space.

---

### Decision · Program_Chairs · 2021-01-07
**Final Decision**

**Decision:**

Reject

**Comment:**

While the reviewer's noted a number of strengths of your paper, the approach that you took, and agreed that you had tackled an important problem, concerns remained about presentation and clarity. I agree. (Here are just a few miscellaneous comments: the very first paragraph of the Introduction needs to be rewritten for clarity, in my opinion. Later on page 1, you use the term "the dynamics of density" but you should not assume that the reader knows what that means. There are typos as well, e.g. "make all predictions base[d] on Equation (6)" on page 4. It would be helpful to know something about why you chose the experimental setups in Synthetic-1, 2, and 3. )

Regarding the similarities between this paper and a previously published article I believe that the authors have addressed these concerns; I hope they are careful to avoid this situation in the future.